# Aerial Bombing Crater Identification: Exploitation of Precise Digital Terrain Models

**Martin Dolejš [1],\*** , **Jan Pacina [2,3]** , **Martin Veselý [4]** and **Dominik Brétt [2]**

[1]    Faculty of Science, J. E. Purkyně University in Ústí nad Labem, Pasteurova 3632/15, 400 96 Ústí nad Labem, Czech Republic

[2]    Faculty of Environment, J. E. Purkyně University in Ústí nad Labem, Pasteurova 3632/15, 400 96 Ústí nad Labem, Czech Republic; jan.pacina@ujep.cz (J.P.); dominik.brett@ujep.cz (D.B.)

[3]    Department of Geomatics, Faculty of Civil Engineering, Czech Technical University in Prague, Thákurova 7, 160 00 Prague, Czech Republic

[4]    Philosophical Faculty, J. E. Purkyně University in Ústí nad Labem, Pasteurova 13 400 96 Ústí nad Labem, Czech Republic; martinvesely.ff@centrum.cz

\*    Correspondence: martin.dolejs@ujep.cz; Tel.: +24-728-959-504

**Abstract:** Places of past conflicts and persistent objects that reflect such events often attract the attention of archaeological prospection which facilitates the construction of conflict narratives. Field prospection as a precise method for localization of aerial bombing craters (as an example of such persistent features) is a highly time- and resource-consuming task. Therefore, methods for automatic identification of such features are evolving. We present a comparison of three methods for possible automatic crater detection based on (a) extraterrestrial crater detection algorithms, (b) geomorphology-based edge extraction, and (c) image pattern recognition via a state-of-the-art convolutional neural network (CNN). All methods were preliminarily tested on a case study of eight Second World War (WWII) aerial bombing crater sites in NW Czechia via Airborne Laser Scanned LiDAR-derived digital terrain models with different spatial resolutions. We found that extraterrestrial crater detection algorithms and geomorphology-based edge extraction methods yield worse results given the standard indices of precision and recall. By comparison, the CNN method utilized for a particular task achieved satisfying results, predominantly with 0.5 m/px resolution (which is often available at the country level) of the input raster. Nevertheless, overall performance with this resolution varies significantly among the sites. Therefore, the quality and readability of the input data are crucial factors for the successful acquisition of precise ordinance location identification.

**Keywords:** aerial bombing; crater localization; UXO; Mask R-CNN; crater detection algorithm; WWII; Czechia

## 1. Introduction

The remnants of bombings by means of exploded and unexploded ordnance have been the focus of scholars for decades. An ongoing effort is underway to research such remnants as features that contribute to the creation of past conflict narratives and help archaeological prospection [1,2]. However, such remnants of past conflicts are a potential source of hazard. The estimation in the case of WWII bombing and other former conflicts indicates a 10–30% [3,4] ordnance failure rate. Given the enormous exposure of particular conflict landscapes to bombing events, the precise localization and identification of such areas are highly relevant for security reasons.

The exact localization of bombing crater fields in situ by field surveying is the most precise identification method. However, this task is highly resource intensive and time-consuming. Therefore,

methods for distant identification using remotely sensed data are evolving. The crater detection studies can be divided according to the input data and methods of crater localization. Archival sources with an exact impact location are scarce but can be found in the form of imagery taken immediately after bombing events to evaluate the bombing precision [2] or in situ localization in detailed (cadastral) maps for damage reporting by local authorities [5]. Other approaches are based on the extraction of features from current remotely sensed data, such as satellite imagery [6] or LiDAR-derived digital terrain models (DTMs) [7–9].

A variety of crater classification methods have been tested. The archaeological prospection of the conflict landscapes backed with the knowledge of the local history could benefit from manual inspection of DTMs [10]. Such an approach serves as a supplementary yet sufficient and useful source for prospecting smaller areas. Moving the scale towards the prospection of larger areas, automatic or semiautomatic methods of extraction are desirable due to increasing time and resource demands. Manual inspection and extraction of aerial bombing craters in an area of 136 km$^2$ with 15,360 features [5] occupied the authors for almost two weeks. The past decade has brought significant improvements in the form of automatic methods, mainly due to developments in machine and deep learning capabilities.

Among the approaches for automatic object detection (inter alia detection of bombing craters), two promising methods—Geographic Object-Based Image Analysis (GEOBIA) and object detection based on convolutional neural networks (CNN)—can be highlighted. The GEOBIA approach builds upon imagery partitioned into discrete nonoverlapping segments and a comparison of a multitude of individual segment properties. This method is regularly used for land use and land cover mapping [11], but a growing number of studies present applications for smaller object detection [12]. Implementation of the GEOBIA approach was tested in the case of detection of charcoal hearths [13], aerial bombing crater detection based on precise satellite imagery in agricultural fields [14], and detection of WWI shell craters [15] based on LiDAR-derived DTM and sky-view-factor hill shading. Regardless of the feature under research, the initial or later phases of identification deal with the edge detection technique for the sake of the division of the object from its surroundings. Several filter types are available for edge recognition and have been applied to the recognition of ancient linear features (Gabor filters, [16]) or feature shape-independent histograms of gradients with applications to craters [14].

In contrast to the GEOBIA approach, CNN methods rely more on the imagery itself. The CNN within the object detection task works with the sliding window technique. The sliding window searches and tests pretrained objects that are held as one class, and the remainder as other classes, dividing the objects into positive and negative images [17]. Object detection via CNN can be implemented using several modified methods, and region proposal-based methods (R-CNN, e.g., Faster R-CNN) are considered among the most accurate [18]. Region proposal-based methods consist of two stages. First, the imagery is partitioned into regions by a region proposal network, and second, potential objects are classified according to the pretrained cropped and labeled images with the presence of objects of interest [17]. R-CNN methods can be divided according to the desired output, whereas the Mask R-CNN method resolves the suitability of the object at the level of pixels; therefore, the results are delivered in the form of a more precise polygon instead of a square bounding box.

The implementation of the CNN in the search for landscape features is evolving, as evidenced by the growing attraction among communities in archaeology and other disciplines outside the field of computer vision-oriented science. CNNs in the search for archaeological objects have been tested in the case of Celtic fields and barrows [19], in addition to roundhouse searches [20] via classification of the objects from LiDAR datasets. Similarly, the CNN method can be applied to precise satellite imagery to highlight potentially valuable sites with better results compared to, e.g., the standalone support vector machine method [21].

Due to the development and precision of CNNs, there are attempts to incorporate such methods into automatic or semiautomatic tools. Concerning the attention to automatization, there are also several tools that are available for crater detection. The development of crater detection algorithms (CDAs) implemented in ready-to-use tools is driven by the prospecting and availability of precise

imagery of the Moon [22,23] or Mars [24,25], facilitating further research on the geology and stratigraphy of planets. Despite the similar morphology of the objects in the study (circular impact craters), the use of automated tools for terrestrial bombing crater detection is not perceived as viable. The factors that can disqualify the tool replicability for terrestrial bombing craters are usually based on the presence of vegetation, erosion of the bombing crater edges, and different sizes of the bombing craters [14]. Such factors lead, inter alia, to testing and development of tools directly on (terrestrial) bombing craters under research (e.g., [26]). Preliminary testing of automatic CDA tools (e.g., PyCDA [23]) for archaeological prospection, in the detection of smaller crater-like objects has been conducted [27]. To our knowledge, however, no effort has been made to assess their applicability to terrestrial bombing craters or the comparability of CDAs with state-of-the-art methods.

Another approach to georelief segmentation is based on the definition and delineation of elementary forms of georelief and their boundaries. Different approaches to this type of georelief segmentation have been introduced by many authors in the literature (pp. 1761–1768, [28]); [29–31]. Minar and Evans [32] proposed a comprehensive approach to the elementarization of georelief. They attempted to connect statistical approaches and basic analytical concepts. The elementarization of the georelief was, in this case, based on the internal homogeneity of individual elementary forms of a georelief, whose boundaries are formed by lines of the morphometric characteristic discontinuity of different orders. In [32], a set of ideal elementary forms of georelief were defined, which were described in terms of [33] using the fitted function. The craters resulting from bombing events may be considered circular elementary forms of georelief [32]. The introduced algorithms [32], including the derivation of high-order morphometrical variables [34], were implemented as a semiautomatic system for the delineation of elementary form boundaries and elementary form recognition using GIS tools and the MATLAB (MathWorks, MA, USA) computing environment. These algorithms have been used only on small test areas smoothed with spline interpolation functions [35].

Given the above information, our paper has three objectives related to the automatization of the aerial bombing crater detection process (or its particular stages) and suitability of the input raster data (spatial resolution). Namely, we (a) evaluate the possible applicability of ready-to-use extraterrestrial CDA tools to terrestrial bombing craters; (b) test the different and (in relevant literature) unpublished geomorphometry-based edge and circularity detection methods; and (c) compare the results with the Mask R-CNN method for bombing crater identification.

## 2. Study Area and Historical Context

For the sake of automatic aerial bombing crater detection testing, we selected eight localities in NW Czechia (Figure 1) that experienced aerial bombing events during WWII. The approximate selection of the bombing sites was conducted via knowledge of historical context and literature (i.e., selection of region or city based on narratives found in archival records). Subsequently, a more precise delimitation was made of the study sites that contain the burden of the bombing via georeferenced imagery taken immediately after the bombing event to verify the success of the attack by allied air forces. Localities B1 and B2 were manually mapped on the ground of archival bombing reports, where individual aerial bombing craters were recorded in cadastral maps and archival aerial imagery, in previous work by the authors [5]; therefore, the exact localization of the aerial bombing craters was taken from this source. For individual archival imagery sources, see Table 1.

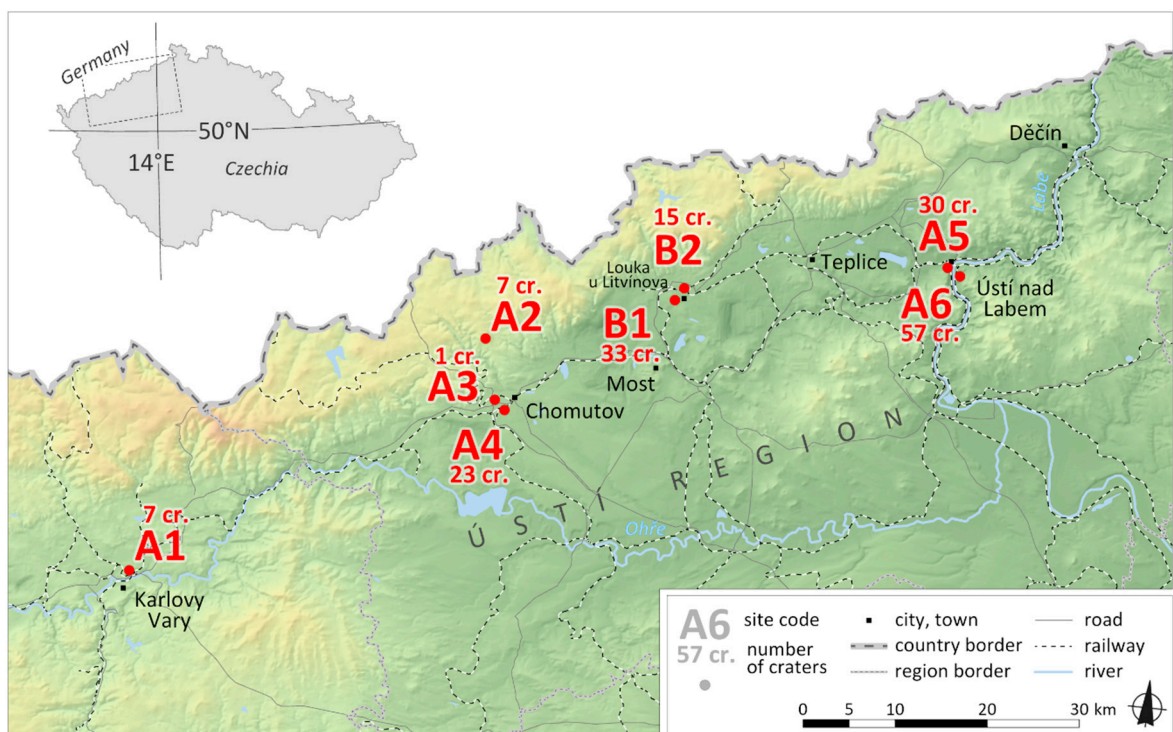

**Figure 1.** Localization of study sites within Czechia.

**Table 1.** Environmental conditions of the study sites, average LiDAR point density, and archival sources (archival imagery collection codes).

| Study Site | Municipality | Mean Altitude (m asl) | Avg. Point Density (m$^2$) | Soil Type | Dominant Land Use (Current) | National Archive London Collection (AIR) no. |
|---|---|---|---|---|---|---|
| A1 | Karlovy Vary | 432 | 12 | pseudogley | forest | 9/406, 29/407. |
| A2 | Blatno | 728 | 29.6 | cambisol | forest/meadows | 9/407. |
| A3 | Chomutov | 352 | 16 | pseudogley | forest | 9/407. |
| A4 | Chomutov | 352 | 19.2 | cambisol | forest | 29/407. |
| A5 | Ústí nad Labem | 251 | 21.4 | cambisol | forest | 29/407, 40/845, 40/846, 40/847. |
| A6 | Ústí nad Labem | 290 | 29.9 | cambisol | forest | 29/407, 40/845, 40/846, 40/847. |
| B1 | Litvínov | 300 | 22.7 | anthrosol | forest | 29/399, 29/386, 29/382, 40/626. |
| B2 | Litvínov | 300 | 14.8 | cambisol | forest/arable | 29/399, 29/386, 29/382, 40/626. |

The abovementioned selection process led to the delimitation of eight localities with similar environmental conditions. As shown in Table 1, the study sites lay mainly at low to moderate altitudes and are covered with common soils, with the exception of the B1 locality, which is partially covered by spoil heaps as a result of mining activities in the region since the Middle Ages. Similar to the soil conditions, almost all of the sites with preserved aerial bombing craters are currently covered with

forests. In addition to the environmental conditions, the historical context and technical details of the bombing are naturally important factors for determining the presence of aerial bombing craters.

The study areas located in NW and W Czechia belonged to Nazi Germany at the beginning of 1938, however, Allied bombing was aimed at the area from May 1944. The target selection of the Allies focused on synthetic fuel production facilities. One of the most prominent hydrogenation factories was situated near Most-Sudetenländische Treibstoffwerke (B1 and B2 study sites). The factory itself and both of the study sites were heavily bombed from 12 May 1944 until March 1945 (for details see Supplement S1). At the beginning of May 1945, the priorities of Allied strikes over NW and W Czechia changed. The attention was refocused on the transportation infrastructure of the enemy, and particularly railways. After the destruction of Dresden (near the NW Czechia border), the key transportation hubs were railway nodes in Ústí nad Labem (study sites A5 and A6) and Chomutov (study sites A3 and A4). Both nodes were bombed in April 1945. Similar to the abovementioned Ústí nad Labem and Chomutov, the railway stations in Karlovy Vary (A1) and Cheb (A2) were affected by the same change of strategy. Although complete information about technical details for a small number of strikes is missing (Supplement S1), we can conclude (concerning aerial bombing crater size and shape) that the bomb types used during the strikes were roughly similar. The most commonly used general-purpose bombs had weights ranging from 250 to 500 lb.

## 3. Materials and Methods

### 3.1. LiDAR Data Acquisition and Processing

The LiDAR data were acquired using a Riegl VUX 1-LR scanner mounted on a small aircraft. The average flight altitude was set to 300 m above ground level, and the strip distance was 200 m (strip side overlap 50%). The following parameters were used: scanner FOV of 80°, pulse repetition rate of 400 kHz, and scan rate of 48 lps. The average point density is presented in Table 1. Localities B1 and B2 were scanned in November 2019, and the remainder were scanned in early May 2020. The A5 and A6 localities had to be rescanned in late June 2020 with repeatedly scanned strips (each line was scanned twice) because the dense vegetation caused holes in the bare-ground point cloud. The data were further processed in PosPAC (Applanix) for trajectory postprocessing and RiWorld (Riegl) for point cloud processing and vegetation filtering. The vegetation filtering was carried out using the Point Classifier tool implemented in RiWorld; the following settings (recommended by the Riegl technical support) were used: Max Object Size: width 30 m, height: 22.4 m; Min Object Size: width: 0.250 m, height: 0.175 m; Outliers below Terrain: 1. The resulting point cloud representing the bare ground was transformed into our national coordinate system S-JTSK (EPSG 5514).

The Riegl VUX 1-LR scanner has a laser beam accuracy of 15 mm and a precision of 10 mm. The on-board Trimble AP20 GNSS-inertial system has a defined (postprocessed) position accuracy (Root Mean Square Error—RMSE) of 0.02–0.05 m. The trajectory was postprocessed in PosPac software with 5 cm positional accuracy. The accuracy of the acquired LiDAR data was tested within the B1 and B2 localities, and 154 control points were surveyed with RTK GNSS with a positional accuracy of 3–8 cm. The processed LiDAR data were compared with the control points, and the RMSE of elevations was 9.5 cm [36], which is sufficient (regarding the spatial extent of the aerial bombing crater features) for our purposes.

The bombing relict identification introduced in this paper was performed by different image-based processing algorithms; thus, the bare ground represented by the processed point cloud had to be converted to an elevation raster. ArcGIS Pro v. 2.5 (ESRI) was used for further LAS dataset processing. The "LAS Dataset to Raster" tool was used for the conversion from the point cloud to the elevation raster. The following settings were used: Interpolation Type: Binning; Cell Assignment: Average; Void Fill Method: Natural Neighbor. This tool allows exporting the elevation raster at defined spatial resolutions; the spatial resolutions of 0.2, 0.5, 1, 1.5, and 2 m/px were used in our research. The average point density (see Table 1) varied from 12 to 29.9 points/m$^2$. The lower values of the average point density

(sites A1, A3, and B2) were caused by the very dense vegetation cover. The highest average point density was achieved at sites A2 and A6. Site A2 has almost no vegetation cover and the LiDAR survey at site A6 was carried out with repeatedly scanned strips. The lowest produced grid resolution 0.2 m/px is not suitable for sites with the lower average point density but will be tested because other sites may benefit from the high spatial resolution.

### 3.2. Visualization Power

To enhance the aerial bombing crater features on derived rasters, we decided to follow the methods previously used for archaeological prospection. Malina [37] introduced methods based on the work of Davis [38]. These methods were used for the enhancement of medieval mining sites, which are very similar to aerial bombing craters. This method works with a combination of three layers derived from the elevation raster. The bottom layer is the classical hillshade visualized in grayscale with 0% transparency. The second layer is the sky-view factor (computed by the Relief Visualization Toolbox, ZRC SAZUre) enhancing the recessed objects—grayscale visualization with 50% transparency. The top layer is the local relief model (LRM) computed according to [38], which is optimal for the enhancement of edges and brake lines, visualized in grayscale with "dark on the top" and transparency 60%. An example of this visualization (at site A6) is shown in Figure 2.

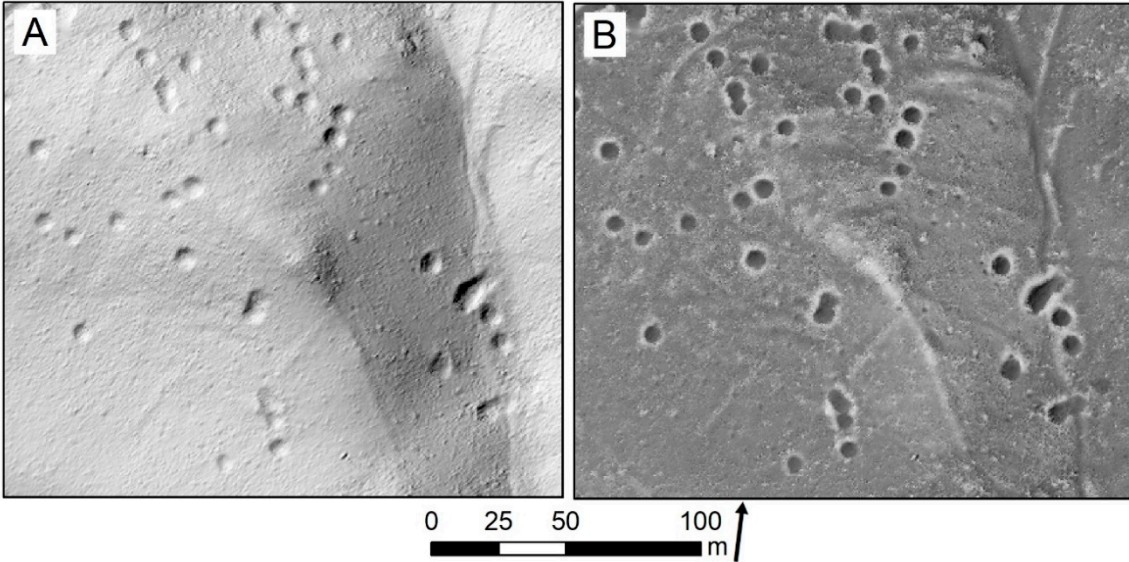

**Figure 2.** Visualization of aerial bombing craters at site A6 (spatial resolution 0.5 m/px) using the hillshade: 315° azimuth illumination and 35° sun elevation (**A**) and the composite of hillshade (315° azimuth illumination and 35° sun elevation, 0% transparency), sky-view factor (10 px search radius, 50% transparency), and the local relief model (60% transparency) (**B**).

### 3.3. PyCDA Algorithm

Considering the possible methods of automated crater delimitation, we added the PyCDA ready-to-use tool ([23] based on [25]) for crater delimitation. The compiled Python tool is based on CNNs and consists of different detectors and classifiers. For the sake of testing, we used a tiny detector as a simplified version of U-Net that provides the likelihood of potential pixels that are identified as aerial bombing crater-related. On the side of classifiers, the 2D convolutional 12 × 12-pixel settings were used. The identified aerial bombing crater model output was significant where the confidence of aerial bombing crater detection was over 90%.

### 3.4. Edge Detectors

Aerial bombing craters can also be seen as specific elementary forms of georelief. The methodological background for delimiting elementary forms of georelief with the use of derived morphometrical variables of higher orders was introduced in [32]. MATLAB software is used to compute partial derivatives of high orders for the derivation of morphometrical variables used for the process of elementary form boundary identification [34]. The boundaries of the elementary forms of georelief (in our case, the aerial bombing craters) are identified as discontinuities in the fields of the derived morphometrical variables [32]. The modified Canny edge detector is used for the analysis [35]. The principle is based on the fact that the Canny edge detector searches for inflex points within the raster of the desired morphometrical variable, and the proposed edges are the local extreme between the inflex points. For more detail about edge detecting algorithms, see Supplement S2.

Two types of derived morphometrical variables of higher orders were selected, based on [32], as the most suitable for identifying aerial bombing crater boundaries: normal change of gradient ($G_n$) and gradient change in the direction of a contour line ($G_t$), as follows (1–3):

$$\frac{\partial z}{\partial x} = z_x; \; \frac{\partial z}{\partial y} = z_x; \; \frac{\partial^2 z}{\partial x^2} = z_{xx}; \dots, \tag{1}$$

$$G_n = \frac{z_x^2 z_{xx} + 2z_x z_y z_{xy} + z_y^2 z_{yy}}{z_x^2 + z_y^2}, \tag{2}$$

$$G_t = \frac{z_x^2 z_{xy} - z_y z_x z_{xx} + z_x z_y z_{yy} - z_y^2 z_{xy}}{z_x^2 + z_y^2}. \tag{3}$$

Whereas the gradient change rasters ($G_n$, $G_t$) define the edge of the possible aerial bombing crater, the craters as features need to be manually identified on input raters. Considering workflow automatization, we extended the process by the Hough Circle Transform [39] for circle delimitation implemented in the Open Source Computer Vision Library (opencv). The algorithm seeks the circular pattern (here the identified edge of the aerial bombing crater) and yields a circle with center coordinates and radius.

### 3.5. Mask Region Based Convolutional Neural Networks (Mask RCNN) Method

As mentioned above, use of the archival imagery data and enhanced DTM resulted in the delimitation of the exact location of individual aerial bombing craters. This aerial bombing crater dataset was divided into training and testing partitions for the sake of training the deep learning model, particularly for the Mask RCNN method.

Locations B1 and B2 (Figure 1) were selected for training with 42 marked aerial bombing craters. Six craters were excluded due to low representativeness (presence of other objects inside the crater, fuzzy edge, or other than circular shape). Therefore, the remaining 125 aerial bombing craters were left for model testing in all six localities labeled "A" in Figure 1.

The selection of a particular CNN model type for smaller object identification followed the pretesting of available model types within the ArcGIS package. The performance of Single Shot Detector, Retina Net, and Mask RCNN were tested with 0.5 px/m spatial resolution leading to selection of Mask RCNN method due to a number of true positive identifications aerial bombing craters.

For the sake of model training, the training raster datasets were divided into image chips with a tile size of 300 pixels with half stride size and structured for the Mask RCNN model. Structured image chips with labels were used for training the models for particular resolutions of the input raster (0.2, 0.5, 1, 1.5, and 2 px/m). We trained our models on the object detection Mask RCNN model with a maximum of 50 epochs with the preconfigured neural network architecture of the ResNet-50 backbone. The resulting delimitation, data preparation, and training were conducted in ArcGIS Pro 2.5.

### 3.6. Validation of Method Performance

For the sake of preliminary validation of the performance of the individual methods (PyCDA, edge detection, and Mask RCNN), we began testing at a spatial resolution of 0.5 m/px to avoid a time-consuming process with no further significant results at other resolutions. The resolution of 0.5 m/px was marked as an optimal solution for the level of noise and readability of the aerial bombing crater features for further preliminary testing of all three methods. The effect of the particular models and raster resolution on the performance was reported in the forms of standard precision and recall (PR) values, the precision–recall graph, and the area under the PR curves (AUC). The precision was calculated as follows:

$$Precision = \frac{TP}{TP + FP}, \tag{4}$$

where *TP* is the true positives (correctly identified aerial bombing craters; resulting modeled polygon respects the shape and area of the initial aerial bombing crater on the input raster) and *FP* is the false positives. Here, false positives are the resulting (model provided) polygons outside the aerial bombing craters on the raster source. The recall values were obtained as follows:

$$Recall = \frac{TP}{TP + FN}, \tag{5}$$

where *FN* stands for false negatives. False negatives were labeled and actually present aerial bombing craters that models were unable to find. Finally, to find the optimal balance between the precision and recall values, we employed the commonly used harmonic mean of precision and recall, i.e., the *F*1 score (6), calculated as follows:

$$F1 = 2 * \frac{Precision * Recall}{Precision + Recall}, \tag{6}$$

The AUC model performance metrics for individual models were calculated together with graphing of the results via the precrec package [40] in R.

## 4. Results

### 4.1. Point Cloud Data Processing and Aerial Bombing Crater Identification

One aim of our study was to test the optimal spatial resolution for automatic aerial bombing crater identification. The "LAS dataset to raster" function implemented in ArcGIS Pro allows interpolation of the input point cloud into a raster with defined spatial resolution. The point cloud processing is described in detail in Section 3.1. Elevation rasters of the following spatial resolutions were produced: 0.2 m, 0.5 m, 1 m, 1.5 m, and 2 m/px. Figure 3 shows the effect of resolution on the details of the resulting raster.

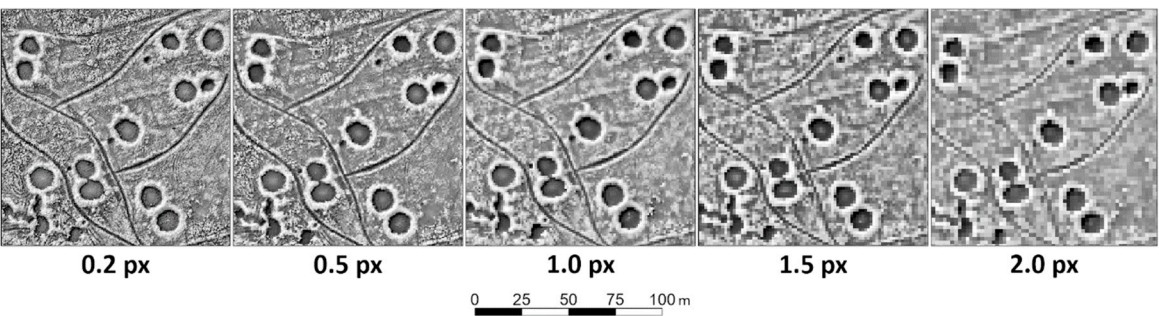

**Figure 3.** Spatial resolution (pixel/meter) and aerial bombing crater detail for individual resolution steps and identical space. Note: the pixel width of the raster varies, from left 600, 240, 120, 80, 60 px.

All of the processed sites were visualized with the combination of hillshade, sky-view factor, and simplified LRM as described in Section 3.2 (see example in Figure 2B). Based on the comparison of the archival imagery sources and our rasters, we manually identified 173 preserved aerial bombing craters together at all locations.

### 4.2. Preliminary Identification and Comparison of Methods

Edge detectors are a possible means of identifying different landform boundaries, including aerial bombing craters in this work. After preliminary testing of 0.5 m/px resolution, three different spatial resolutions were tested—0.2, 0.5, and 1 m/px—to check for possibly better results. The edge detector method is very sensitive to any noise in the input data; thus, the data were smoothed with a low-pass filter before analysis. The function Filter implemented in ArcGIS Pro was used with the "low-pass" option (using a 3 × 3 cell neighborhood to compute the average (mean) values from the neighboring cells).

The 0.5 m/px resolution raster was used for further boundary delimitation. The 0.2 m/px raster contains too much terrain detail, and the resulting boundaries are hard to identify. Conversely, the 1 m raster is too rough, and some boundaries are lost (Figure 4B–D). The significance of the delineated edges is defined by the assigned weight value. The boundaries above the defined weight value for each site were extracted and further used for aerial bombing crater delineation using the Hough circle transform, as shown in Figure 4C.

Regarding aerial bombing crater identification, we found significant differences among study sites. Testing the precision and recall achieved moderate success in localities B1 and A6, where aerial bombing craters are easily differentiable from other features (built-up areas, ponds, etc.) present on the raster and the features itself are approximately regular (circular). Contrary to these two study sites, the remaining six performed poorly with low or no correctly identified aerial bombing craters and a high volume of false positives. Therefore, the overall values of precision and recall indices (Table 2) are low and, generally, we can label the overall identification for all study sites as poor.

**Table 2.** Validation indices of the identification test for individual methods.

| Model | Spatial Resolution (m/px) | Recall | Precision | F1 Score |
|---|---|---|---|---|
| PyCDA | 0.5 | 0.07 | 0.03 | 0.05 |
| Edge Detection | 0.5 | 0.23 | 0.17 | 0.19 |
| Mask RCNN | 0.5 | 0.74 | 0.58 | 0.65 |

A similar situation with input raster suitability was found in the case of PyCDA, where 0.5 m resolution was used for preliminary testing. Nevertheless, the preliminary test yields very low precision values (Table 2); therefore, we did not continue to test other resolutions. In other words, the resulting identification placed among correctly labeled aerial bombing craters a high number of incorrectly labeled aerial bombing craters (false positives) in certain localities (surprisingly, at the B1 study site, where identification based on edge detection performs well). The second issue was raised with the localities with a lower density of aerial bombing craters and the presence of similarly shaped features where the method correctly detected one or no features with the equivalent number of false positives ("A" labeled study sites). By comparing the PyCDA and edge detection method by the F1 value, we found the edge-based approach to be more precise, although, both of the methods can be considered as unreliable.

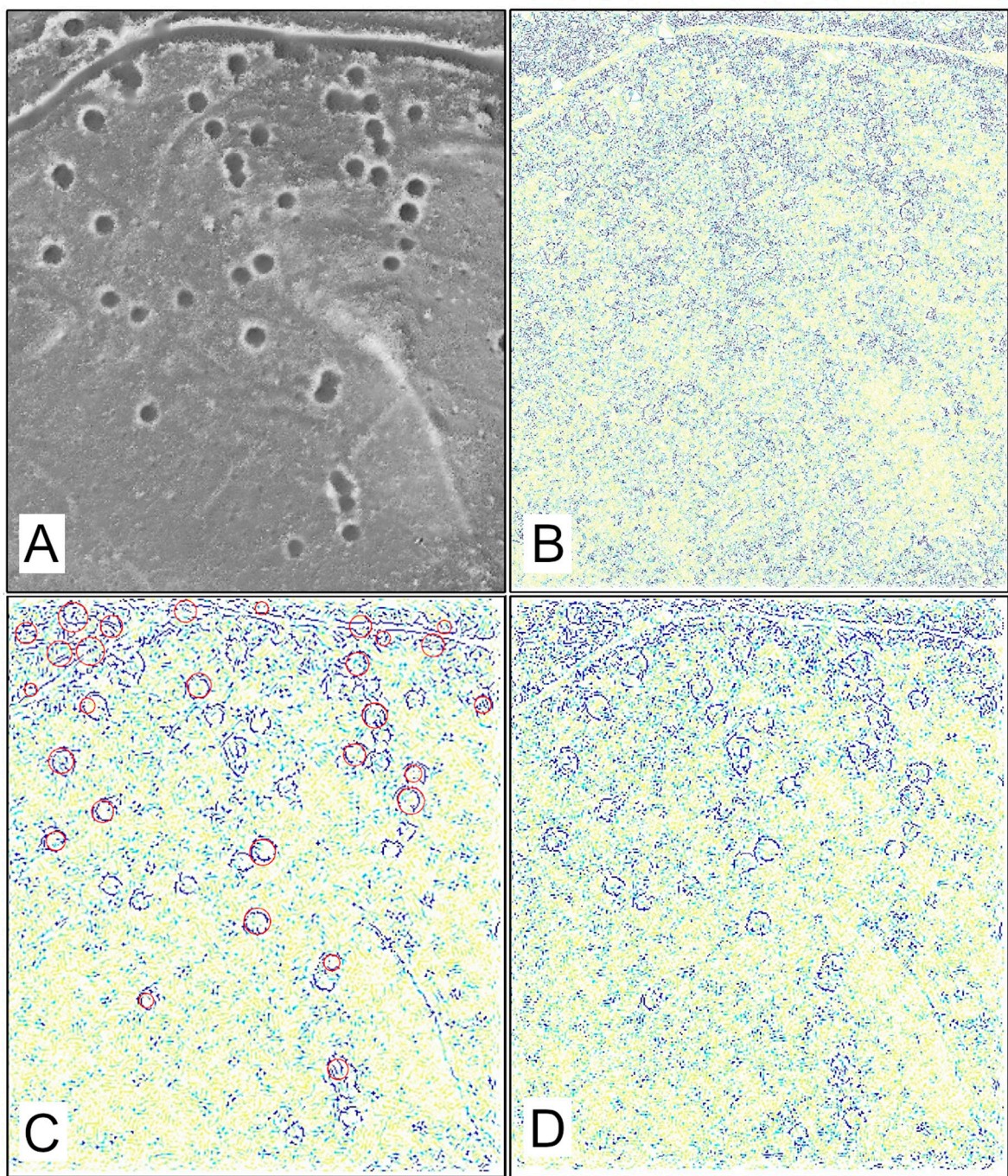

**Figure 4.** Aerial bombing crater boundaries delineated using the edge detectors. Dark blue is the most significant edge, yellow is the least significant edge. (**A**) The input surface visualized using image fusion (see Figure 2 caption). (**B**) Edges delineated from a 20 cm raster. (**C**) Edges as delineated from a 50 cm raster with marked circles (red) based on the Hough circle transform. (**D**) Edges as delineated from a 1 m raster.

Better results concerning all indices (Table 2) were found while moving to identification based on image recognition and training on a subset of the rasters (Mask RCNN). Preliminary testing on 0.5 provided the best results among all three methods; therefore, we proceeded and tested all input resolutions (0.2, 0.5, 1, 1.5, 2 m/px).

### 4.3. Effect of the Spatial Resolution of Input Raster—Mask RCNN

Examination of the suitability of the individual spatial resolution of the input raster for recognition of the aerial bombing crater features led to the construction of the precision–recall curve. As shown in Figure 5A, the best precision–recall performance yielded a 0.5 m/px resolution (the optimal performance would be a line with an intercept value of 1 on both axes of the graph). The worst results were found using 1.5 m/px resolution. The relevance of the model can be expressed as a single value of area under the curve (AUC, Table 3). Corresponding to the graph, the best performance was not obtained with the most detailed resolution (0.2 m/px). The expected decline of the performance after 0.5 m/px resolution was not observed. Due to a lower number of false positive results, the resolution of 2 m/px yields better AUC values than those of the 1 and 1.5 m/px resolutions. Contradictory to the output of the AUC, the overall F1 score in Table 3 exhibits trend-like behavior with a peak at 0.5 m/px resolution.

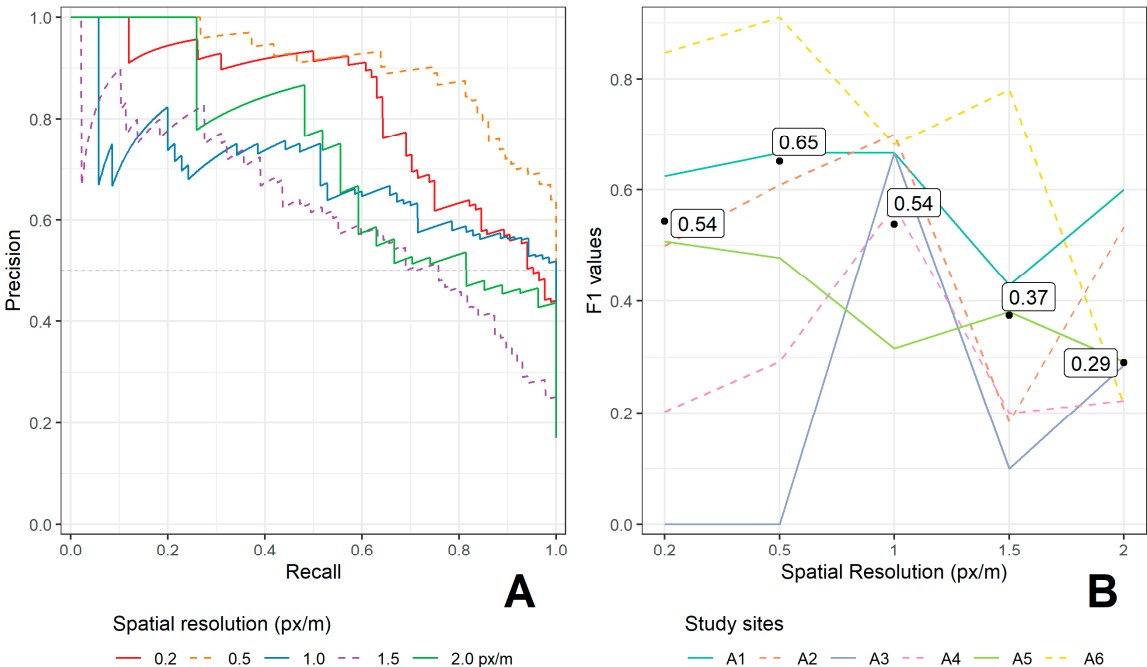

**Figure 5.** Mask RCNN relevance indices. (**A**) Precision–recall curve for individual spatial resolutions. (**B**) Harmonic mean of precision and recall values (F1 score) for all study sites and spatial resolutions. Black points with labels represent F1 scores for merged values for all study sites and individual spatial resolution (F1 score in Table 3).

**Table 3.** Reliability indices for individual spatial resolutions.

| Spatial Resolution (m/px) | Recall | Precision | F1 Score | AUC |
|---|---|---|---|---|
| 0.2 | 0.71 | 0.44 | 0.54 | 0.82 |
| 0.5 | 0.74 | 0.58 | 0.65 | 0.91 |
| 1 | 0.57 | 0.51 | 0.54 | 0.69 |
| 1.5 | 0.74 | 0.25 | 0.37 | 0.62 |
| 2 | 0.22 | 0.42 | 0.29 | 0.73 |

A closer look at the identification results for particular study sites and resolutions, similar to the merged results for all study sites, yields no clear trend of, e.g., decrease in F1 value together with spatial resolution. As shown in Figure 5B, at the 0.5 m/px resolution the overall good result is driven by the performance for study site A6, which contains a higher volume of well-preserved aerial bombing craters. Other site-specific resolution results are present at 1 m/px, where most of the study

sites perform relatively well, with the exception of site A5. Similar to the previous resolution, the worst overall results are driven by performance for a single site. In other words, we observed significant differences in the behavior of individual study sites and spatial resolutions.

## 5. Discussion and Conclusions

To contribute to the crater identification issue, we presented results from three different methods based on recognition of aerial bombing crater features on a custom LiDAR-derived DTM. Due to the different methods and input raster spatial resolutions used, causes of the variation of the results are therefore twofold. First, the results vary among methods from unsuitable to good, and second, we found variation based on raster spatial resolution.

First, we tackled the issues that could influence the identification process, i.e., mainly input data applicability and possible variation of aerial bombing crater features. As shown for the identification success rate among study sites (Figure 5B), the aerial bombing crater readability for particular methods varies. This variation could be a result of the aerial bombing crater conditions. In this study, we observed better results with clear circular shapes and easily differentiable rims of the feature. Worse conditions for feature identification, in contrast, were displays of vegetation overgrowth, the presence of material inside the crater, and rim erosion. Such conditions are later propagated into the applicability of the LiDAR-derived DTM and make craters difficult to identify. In addition to the condition of the aerial bombing craters themselves, the LiDAR data density influences the quality of the derived DTM. The density of LiDAR points varies among the study sites; individual sites A1 and A3 display lower average point density than recommended for 0.2 m/px DTM. Despite this fact, we did not observe a clear dependency between the success of the identification (expressed as the F1 score, Figure 5B) and average point density (c.f., A1 and A2 study site results) However, this fact is influenced by the limited number of study sites and variation of aerial bombing crater numbers at individual study sites.

Another minor issue arises with the aerial bombing crater morphology variation itself. As previously discussed, Allied airstrikes carried bombs with different weights and resulting blast damage. Although aerial bombing crater morphology is influenced by other factors, such as pedological conditions and fuse type, the variation of aerial bombing crater diameters inflicted by 250 and 500 lb. bombs ranges from one to two meters [41]. The theoretical variation in crater size implicated by various bomb weights was verified in our samples in which the deviation in size was low. Such a difference in size therefore should not be an issue for CNN-based methods, and even less for edge detection methods, in which the size of the crater does not pose a significant issue for delimitation due to the following scale-independent recognition of circular shapes.

Despite the possibility of the application of ready-to-use extraterrestrial tools [23], i.e., PyCDA for terrestrial object detection [27], our results exhibit low precision and recall values. In other words, the identification of terrestrial craters resulted in a large number of false positives. It is highly probable that the poor results can be attributed to differences in the morphology and size of the extraterrestrial craters [14]. Although the tool is trained on lunar craters with generally larger (up to subkilometer) sizes and sharp rims, terrestrial craters, even in the case of highly precise LiDAR-derived DTM, usually do not display such properties. Other sources of error could be rooted in the type of raster employed in the testing. We preprocessed the raster to enhance the terrain morphology, whereas the lunar crater tools are trained on a simple hillshade raster (c.f., Figure 2).

Edge detection based on local geomorphology yielded more precise results compared to the use of an extraterrestrial crater detector. However, identification success varied significantly among study sites. Such discrepancy could again be explained by the quality of the input data (aerial bombing crater readability) for particular sites and the wide purpose of the algorithm. The edge detection employed in our methodology exhibits good results in the case of large-extent feature identification (e.g., watersheds). Regarding the detailed scale of aerial bombing crater features, the level of noise increases and makes consecutive recognition more difficult. The overall comparison of the methods

(Table 2) yielded better indices of F1, precision, and recall for the Mask RCNN method. These better results were not influenced by an unbalanced number of testing sites, where PyCDA and an edge detector were tested on two sites (B1, B2) in the case of Mask RCNN for training. Both methods scored above average on these sites, thus improving the overall results.

The best performance according to precision, recall, and F1 score was observed with the Mask RCNN method with 0.5 m/px spatial resolution of input raster datasets. The abovementioned indices of model relevance are comparable to results from other studies focused on bombing crater identification. Similarly, the CNN approach was tested on a larger sample [14] and led the authors to better precision and slightly worse recall values. Using the GEOBIA approach, bombing crater identification on precise satellite imagery in visible spectra [14] with a large-scale sample led to better F1 results (0.79; here, the best performing scenario resulted in 0.65). The estimation of ground truth at sampling sites in the case of a similar approach [15] led to a success rate of 84% (here, 58%).

Detailed testing with different spatial resolutions resulted in peak values of the F1 score for the selection of the 0.5 m/px resolution. In addition to the employed method for identification, the 0.5 m/px raster exhibited the clearest readability for manual interpretation following raster hillshade manipulation. The declining trend of the F1 score towards the coarse 2 m/px resolution was partially observed after the 0.5 m/px resolution. Similarly, the more precise 0.2 m/px does not yield better results, possibly due to a higher volume of noise caused by a higher degree of terrain detail and, more importantly, by lower than recommended average point density. Despite this fact, our finding of the most suitable 0.5 m/px resolution implies the possible replication of the method for other areas. Because the LiDAR dataset at a resolution of 0.5 m/px or better is available for entire countries (e.g., Czechia, Denmark, UK), there should be no further need for custom LiDAR preparation for replication in other regions that potentially carry the burden of past conflicts.

However, during testing, shortcomings of the Mask RCNN approach were identified. In particular, the input raster should incorporate convexity or concavity information of the features when compared with the method of labeled convex features with similar size (here, false positives). We believe that further preprocessing adjustment of the input raster data (augmentation based on concavity and convexity as a color difference) will lead to an increase in precision and recall.

**Supplementary Materials:** The following are available online at http://www.mdpi.com/2220-9964/9/12/713/s1, Supplement S1: Supplement1—Airstrikes list; S2: Supplement2—The edge detecting algorithms in detail; Data: data used for this article are available as Mendeley Datasets doi:10.17632/2g69hgrpvn.1.

**Author Contributions:** Conceptualization, Martin Dolejš and Jan Pacina; Methodology, Martin Dolejš and Jan Pacina; Validation, Martin Dolejš; Formal Analysis, Martin Dolejš and Jan Pacina; Resources, Jan Pacina, Martin Veselý and Dominik Brétt; Data Curation, Jan Pacina, Martin Veselý, Martin Dolejš; Writing-Original Draft Preparation, Martin Dolejš and Jan Pacina; Writing-Review & Editing, Martin Dolejš, Jan Pacina and Martin Veselý; Funding Acquisition, Jan Pacina. All authors have read and agreed to the published version of the manuscript.

**Funding:** This work was supported by Project No.: UJEP-SGS-2019-44-001-3 (Spatial data remote sensing methods as a tool for modelling and observing the environment) covered by student grant competition at UJEP—Jan Evangelista Purkyně University in Ústí nad Labem.

**Conflicts of Interest:** The authors declare no conflict of interest. The funders had no role in the design of the study; in the collection, analyses, or interpretation of data; in the writing of the manuscript, or in the decision to publish the results.

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
