# Peer review of "Aerial Bombing Crater Identification: Exploitation of Precise Digital Terrain Models"

_ijgi, doi:10.3390/ijgi9120713_

Round 1
Reviewer 1 Report
Bombing crater identification: exploitation of LiDAR-based DTM
(numbers refer to numbered lines)
3: terminology - use “LiDAR-derived” throughout (as used on p2).
19-20: the connection between the mapping of explosion craters and the search for unexploded explosive bodies is indirect, at best and the article makes no attempt at substantiating it; it should be removed from the abstract.
20-21, 45-73, and throughout the entire text: terminology - use the term “crater” with utmost precision (e.g. air bomb explosion craters).
But in that case, how are they related to unexploded air bombs?
53: include other relevant bibliography, e.g. DOI: 10.1002/esp.4586 and DOI: 10.1016/j.apgeog.2013.10.004, etc.
57-58: Can you substantiate the claim? An expert can manually map at least a thousand explosion craters per day. Perhaps add an estimate, based on your work, how many explosion craters have been identified per person/day?
136: briefly in a few words describe the method used in /5/ (e.g. historical maps)
174: remove “, Canada”
175: remove “, Austria”
185 - 189: point cloud processing must be described in more details, i.e. similar to the raw data acquisition and processing (name algorithm and filter, e.g. PTIN/Axellson 1999, Lastools’ LASground, settings …; interpolation used).
188-189: also report on average ground point density (if 20 points/m2 are acquired in forested area the expected ground point density suffices for 0.5 m/px, but 0.2 m/px makes no sense); this is very important because id directly pertains to your conclusions.
170 and 189, 281: terminology - unify “points/m2” (whole word) “m/px” (shortened), “m/pixel” and similar throughout the text.
196: remove “, Slovenia” and all other mentions of countries throughout the text.
196 - 200: report on settings used for each visualisation according to the accepted standards (e.g. according to DOI: 10.3986/9789612549848)
198 and 204: which method was used, LRM or simplified LRM? Despite the similar name they are two different visualisation methods.
253: either remove the mention of hardware (preferably) or describe the entire hardware setup if it is pertaining to the content (e.g. to report on the computing time).
279: terminology - change “converting” to “interpolating”
278 - 282: see 188-189; in addition, the interpolation used (nearest neighbour) is ill advised for this purpose. At this point there is not much to be done, since the entire analysis is based on the suboptimal input, which was caused by suboptimal interpolation.
295: such images must be produced at 100% crop size of the densest image (i.e. 1 px in image is 0.2m for all). This must also be reflected in the figure caption.
283-288: this should be moved to section 2.1.
301: describe the algorithm and setting for the filter used.
320: Figure 4 most likely
321-322: change “composite visualization” to image fusion (see DOI 10.3390/rs11070747 or similar)
368 - only use “digital terrain model (DTM)” when first mentioned, hencefort “DTM” sufices.
371: see 188-189
378-379: false statement, the preservation of an archaeological feature has nothing in common with the airborne LiDAR data quality. However, poor preservation of archaeological features remains key area of differentiation between the “manual” and “automatic” feature detection in archaeology.
Author Response
Dear Sir/Madame,
please find the revisited manuscript with our reactions on your remark attached as a single pdf file.
On behalf of authors,
Martin Dolejš

Reviewer 2 Report
This paper compares three different approaches to crater detection, with interesting results. This reviewer thinks that the paper is well structured and the methods were clearly explained. Below several remarks in detail: 1. Is the term "exoterrestrial" standard? I have not been able to find this word in the dictionaries/journals... 2. The results clearly show that M-RCNN gives the best results, however I think in both the pyCDA and Matlab-based edge detection the authors didn't elaborate enough; e.g. why use the Canny edge detector and not another? Did the authors test different parameters to acquire this result? I think one of the advantages of algorithmic approaches such as edge detection when compared against deep learning based methods, is that we have the possibility to tune the parameters better. It would be better for the paper if the authors can explain in more detail their experience in using these algorithms too. 3. On a similar note, more details should be added regarding the deep learning method used. Granted the authors seem to rely on ArcGIS' implementation of M-RCNN, but I would have preferred to have a little more in-depth explanation. 4. Concerning the title: I'm not sure the title describes the paper best, since here lidar is only used as a source of data and in fact the authors do not explore in too much detail the geometric aspects of lidar except the resolution. I suggest modifying the title a little bit to better reflect the contents of the paper, which for me is more on the domain of image processing than lidar processing... 5. English is good, but typos and minor grammatical errors are still aboundAuthor Response
Dear Sir/Madame,
please find the revisited manuscript with our reactions on your remark attached as a single pdf file.
On behalf of authors,
Martin Dolejš
